# The Impact of E-Book Reading on Young Children’s Emergent Literacy Skills: An Analytical Review

**DOI:** 10.3390/ijerph18126510

**Published:** 2021-06-16

**Authors:** Carmen López-Escribano, Susana Valverde-Montesino, Verónica García-Ortega

**Affiliations:** Department of Research and Psychology in Education, Faculty of Education—Teaching and Learning Centre, Complutense University of Madrid, 28040 Madrid, Spain; svalverd@ucm.es (S.V.-M.); veronicagarciaortega@ucm.es (V.G.-O.)

**Keywords:** reading, e-book, childhood, literacy, preschool, first grade, phonological awareness, vocabulary

## Abstract

Young children’s use of digital devices is increasing as we progress through the 21st century and handheld and mobile devices, such as smartphones and tablets, have become increasingly available. While older children using tablets to read has been more broadly investigated, less is known about the impacts of digital reading on children at the stage of literacy acquisition. An analytical review was conducted on the effects of interactive e-book interventions for young children’s literacy development when compared to (a) listening to print books, (b) regular school programs, and (c) reading non-enhanced and non-interactive e-books. A significant additional beneficial effect of e-book interventions was found for phonological awareness and vocabulary learning based on data from 1138 children in 14 randomized controlled trial (RCT) studies. When e-books are properly selected and used, children develop literacy skills equally well and sometimes better than with print books. Additionally, e-book interventions outperformed the regular school program in the development of literacy skills. Similarly, enhanced e-book conditions revealed benefits over the non-enhanced e-book interventions in literacy skill acquisition. The impact of these findings related to health issues, e-book design, disadvantaged populations, and adult-led e-book sharing is discussed.

## 1. Introduction

Children are growing up in a digital media environment where interactions with digital media are an increasing part of children’s daily lives in classrooms and at home. More children, across all levels of society, are using interactive and mobile media on a daily basis [1]. In a recent survey of parents of children aged 8 years and under, the majority (98%) reported that they live in a home with some type of mobile device [2]. As a result of this exposure to technology, children today have many opportunities to explore digital devices and play with them. Many activities in children’s lives are digital, including early literacy experiences. Children’s books are increasingly available in a digital format on electronic devices—often handheld and mobile [3].

E-books present interactive multimodal information as written text, oral reading, music, illustrations, animations, and hotspots that are activated by touching or pressing the touch screen to generate sound and animation (see an example of the e-book “A Shiver of Sharks”: http://bit.ly/2nM3Gr8) (accessed on 15 April 2021) [4]. With digitization, new opportunities for the mediation of multimodal text have emerged. Among the potential advantages of e-books is that they are easily accessible and interactive for beginning readers who cannot yet decode text or are just beginning to learn to decode. Even children with emergent literacy skills who cannot yet read can explore e-books by themselves without the help of an adult [5]. This invites reading practices that may differ from traditional book reading, due to the affordances of the digital touch screen and the social settings in which it is used [6].

However, questions regarding young children engaging with e-books arise regarding whether these digital stories are as beneficial for children as print books read by an adult. There has been much hope for the educational potential of interactive media, such as e-books, along with fear about their overuse during early childhood, a period of rapid brain development. We are only beginning to understand what this digital shift means for young children’s early literacy development [3]. The following questions are still unresolved: What do e-books bring to a child’s early literacy experience? Additionally, what are the digital reading potentials for improving and enriching the literacy environment of a young child?

### 1.1. Effects of Digital Reading on Children’s Literacy Skills

In the last decade, five reviews [7,8,9,10,11] and four meta-analyses [12,13,14,15] have been carried out to compare children’s reading acquisition ability when using digital devices versus the use of traditional printed books [7,8,9,10,11,12,13,14].

Two of these reviews [8,9] concluded that e-books and printed books play different roles in learning to read and, therefore, the false dichotomy between these two forms of reading should be eliminated, because they are different experiences of reading. A review investigating the role of digital reading [9] found that tablets may improve emergent literacy skills. However, parent or teacher scaffolding is needed to maximize the benefits of e-books. A different review [10] reported that well-designed e-books are as effective as printed books in improving reading acquisition outcomes. A meta-analysis on multimedia stories, which compared independent e-book reading with traditional shared book reading, found that multimedia features can provide similar scaffolding to reading with an adult [12]. A different meta-analysis [13] found that stories presented through multimedia can support and even strengthen children’s understanding of the story compared to listening to stories in more traditional settings, such as storybook reading. Similarly, a research synthesis of experimental and quasi-experimental studies investigating the effects of e-books on children’s literacy development found moderate effects of e-books on reading comprehension [14]. However, a recently published meta-analysis examining the effects of e-book use on literacy outcomes found no statistically significant effects between e-book and non-e-book conditions on norm-referenced standardized test measures of reading and reading comprehension [15].

Despite the positive outcomes of e-book reading, a number of researchers have taken a more critical view on e-books due to their incorporation of features such as hotspots that contain animation, sound, and other multimedia effects and may distract young readers from the story content and negatively affect their understandings of the story’s main theme [10,11,13,14]. Some hotspots are congruent with the story (i.e., support the story’s content) and others are incongruent (i.e., they do not align with the story content and might even distract the reader from it) [4]. In this sense, a recent analysis on e-book design reveals that the first published digital books included hotspots that often had little or no relevance to the story and distracted children from language and literacy learning. Nevertheless, important improvements have been made in e-book design compared to former years as the number of interactive visuals and of hotspots seem much lower than in previous years, and they are more congruent; that is, they elaborate or extend the story line, as it is advisable [16].

Another finding of the literature on e-book reading is that adult–child interaction and e-book sharing with young children differs from sharing print books [8,10,11]. Parents reported their children not only read traditional books more than electronic books, but enjoyed them more and paid more attention to them. Caregivers also reported participating in more talk about the story when reading print books than electronic books [17]. Similarly, teachers sharing an enhanced digital book struggle to define their role [6]. When sharing a digital book, children may be occupied by the interactive elements in the book (tapping hotspots initiates sounds, simple animations, and dialogue/sounds from the characters) while ignoring the story. More research is needed to further explore new routines that develop when families or educators have access to a set of well-designed digital picture books.

Despite these concerns, previous empirical studies have identified the effects that e-books can have on the development of children’s literacy skills. Due to their many unique features, e-books provide children with many opportunities for promoting their emergent literacy skills. For instance, studies have shown that digital books support the development of children’s print and phonological awareness [18,19,20], vocabulary development [18,21,22], spelling development [23], and reading comprehension [5,24]. These skills (e.g., phonological awareness, print awareness, vocabulary, and reading comprehension) are considered significant to the development of children’s emergent literacy abilities.

### 1.2. Health and Developmental Concerns

Heavy media use during preschool years is associated with negative effects on children’s health, general development, and outdoor play [1]. The risks of children spending a lot of time in front of a screen have been well documented by research: Addiction [25], obesity [26], negative effects on motor dexterity [27], and eye fatigue [28], among others.

Moreover, since the cognitive control mechanisms are still immature in young children [29] the high exposure to digital games (also found in e-books) makes them especially vulnerable to develop pathological gaming behavior [30]. Pathological gaming has become a major concern for health care professionals during the last years [31] and has been included as a game disorder in the International Classification of Disease (ICD-11) Manual [32], as well as, a condition called Internet Gaming disorder in the Diagnostic and Statistical Manual of Mental Disorders 5 (DSM-5) [33]. Thus, evidence is sufficient to recommend time limitations on digital media use for children 2 to 5 years to no more than 1 h per day [34].

It is reasonable for parents and teachers to be concerned about the excessive use of digital content, especially in young children. However, touch screen devices are rapidly gaining place in the lives of families with young children, and parents also hold positive views toward technology use and are able to identify a range of benefits that their children have acquired [1,2]. Today, media represent just another environment; children do the same things they have always done, only virtually.

A different issue related to health that has changed education dramatically and globally, with the distinctive rise of e-learning, has been the COVID-19 pandemic.

Considering these two scenarios, young children growing up in contexts saturated with technology and the shift away from the classroom and the adoption of online and digital learning in many parts of the world, policies and recommendations must evolve and provide thoughtful, practical advice to parents and teachers founded on evidence, and not based merely on the precautionary principle [35].

### 1.3. The Present Study

The current review is an attempt to better understand the effects of e-book use on reading outcomes for children aged 7 years or younger. This specific age group was chosen based on previous research findings, suggesting that book reading in the first 6 years of life is related to emergent reading and reading achievement [36].

Previous reviews [7,8,9,10,11] and meta-analyses [12,13,14,15] have broken ground into furthering our understanding of how e-books can enhance literacy skills and practice. However, none of these reviews have been conducted for investigating the effects of e-book use on the reading outcomes of preschool, kindergarten, and first grade students in the last 10 years. There is a lack of knowledge concerning digital reading with regard to the extent to which e-book interventions support different aspects of a student’s emergent literacy acquisition and to which enhanced features within e-books improve literacy outcomes. There is a pressing tension between the actions that could be taken and the actions that should be taken with e-books [14].

Given these gaps in our understanding of the extent to which e-books support early emergent reading skills, this study is dedicated to e-book-based interventions for the acquisition of literacy skills of children from 4 to 7 years old, including studies from 2010 to 2020. We consider an e-book as any form of electronic text that contains key features of print books. This synthesis includes both traditional e-books (static digital version of a print book) and enhanced e-books (with the key features of a traditional print book, but with animated and interactive content, audio, and video). This definition of an e-book excludes other software that can support literacy development, but is not embedded in a book format.

The aims of this paper are to (a) explore how effective e-books are for supporting early literacy skills: Concepts about print, phonological awareness, and vocabulary as well as reading comprehension; (b) examine the effects of e-book reading interventions compared to printed book reading; (c) study the results of enhanced e-book reading interventions compared to non-enhanced e-book reading interventions or simple static e-books; (d) study the effects of e-book reading interventions compared to the regular school program; and (e) analyze the reviewed studies from adult support, e-book design, and disadvantaged populations. We were specifically interested in the additional effect on literacy development of interactive or enhanced e-book interventions as compared to the more traditional presentation of stories, such as reading a print story book, following the regular literacy curriculum, or reading a more traditional e-book without enhanced characteristics.

The present analytical review used more stringent inclusion criteria than previous reviews and meta-analyses, only including randomized controlled trial studies (RCTs) that have a control group and a pre- and post-intervention design, including baseline measures. Meta-analytic techniques to control for the effects of statistical dependence on effect sizes were used. However, due to the scarce number of RCTs on this topic and the differences in methodologies and excess of variables involved in the reviewed papers, it was decided to exclude meta-analysis procedures. However, the present study could be used as a precursor to a meta-analysis.

A total of 14 studies met the inclusion criteria, and throughout these studies, nine conditions compared e-books to print books, ten conditions compared enhanced e-books to non-enhanced e-books, and seven conditions compared e-books to the regular literacy curriculum.

## 2. Materials and Methods

In order to conduct the present review, we have followed the methodological approach to systematic reviews by Hammersley [37] where exhaustive searching for relevant material, transparent methodological assessment of studies, and a formal and explicit method for the synthesis of findings, have to be explicitly established.

### 2.1. Data Collection

The following criteria were used to assess the eligibility of the studies:A randomized controlled trial (RCT) design and matching procedures to ensure equivalence at baseline for treatment and comparison groups were used. These features of study design are critical to establishing the internal validity of study findings [38];The study contained a targeted e-book reading intervention. This included interventions that used e-book reading to target specific literacy skills; it also included interventions with kindergarten, preschool, and first grade age children; interventions delivered by researchers, teachers, or parents; and interventions delivered on a one-to-one basis or to children in groups;The study measures’ dependent variables that focused on literacy outcomes (e.g., concepts about print, phonological awareness, vocabulary, and reading comprehension);The study contained at least one literacy outcome measure that yielded an objective quantitative score. These outcome measures included mainly scores on author-designed assessments, such as testing children’s knowledge of the vocabulary to which they had been exposed in the course of intervention;The study contained at least one control group or different treatment conditions. This included studies with passive or “business as usual” control groups and studies with active control groups, including control groups exposed to print book reading and to non-enhanced conditions of e-book reading;The participants in the study were children aged 7 years or younger, regardless of language, country income level, race, or other sociodemographic indicators;Sufficient empirical information to calculate effect sizes was provided.

We searched for relevant published articles from January 2010 to January 2021. Literature searches were conducted in 9 databases: ERIC, Education Database, Psychology Database, Google Scholar database, PsycINFO, PsycARTICLES, Medline, PubMed, and Social Science Database. Search terms included the following: electronic book, digital book, e-book, multimedia book and literacy, reading, reading comprehension, vocabulary, phonological awareness, and random controlled trial. Combinations of these terms were used to search for the targeted articles.

The initial search yielded 2746 studies. After importing abstracts using Mendeley and removing duplicates, 1978 studies remained; after reviewing the abstracts, 106 studies were selected as possible studies to be included. Next, authors conducted a full-text review of the 106 remaining articles, resulting in a final sample of 14 studies that met the inclusion criteria. A PRISMA flowchart is shown in Figure 1 [39].

### 2.2. Coding Procedures

The characteristics of the 14 reviewed studies are presented in Table 1, Table 2 and Table 3. Table 1 presents the conditions comparing exposure to e-books and printed books; Table 2 shows comparisons between different types and conditions of e-books; and Table 3 shows comparison between exposure to e-books and to the regular school program. The studies were coded by the three authors of the present study; 94% interrater agreement was achieved, and discrepancies in coding were resolved via discussion. We included general codes of the studies (e.g., author, study design, and participant information), as well as specific codes for e-book use (e.g., intervention description and device used to access the e-book, number and type of e-books, intervention duration, etc.).

### 2.3. Calculation of Effect Sizes (ES)

All 14 studies included in this review provided adequate statistical information (e.g., the pre- and post-intervention control and treatment group means, standard deviations, and numbers of participants in each group) that made it possible to calculate the effect sizes for the differences among different intervention, control, or treatment groups on literacy outcomes (e.g., concepts about print, phonological awareness, vocabulary, and reading comprehension). We applied the effect size formula based on the mean pre–post change in the treatment group minus the mean pre–post change in the control group (or a different treatment condition), divided by the pooled pretest standard deviation [40]. In order to reduce the bias estimation of effect sizes, a bias correction parameter, which considered the sample size of the two groups, was also applied, as recommended in [52]. This formula was chosen because (a) it is recommended for studies with repeated measurements in both the treatment and control groups, as is the case of the present study and (b) in terms of bias, precision, and robustness to heterogeneity of variance, it is favored over alternate effect size estimates [46]. Based on recommendations, estimates of less than 0.20 were considered non-significant; estimates of 0.20–0.49 were considered small effects; 0.50–0.79, medium effects; and values above 0.80, large effects [53].

## 3. Results

Overall, 14 studies met the inclusion criteria. A total of 11 studies were conducted in the last five years (i.e., 2014–2018). The remaining were published between 2011 and 2013. Studies were classified based on (1) e-books versus printed books (Table 1 and Table 2), enhanced e-book versus non-enhanced e-book (i.e., interactive versus static e-books), (Table 2), and (2) e-book versus the regular school program (Table 3). Some of the selected studies contributed to more than one of these categories. The reading outcomes were classified into four variables: Concepts about print, phonological awareness (including measures of letter sound and letter name), vocabulary (including measures of receptive and expressive vocabulary, as well as other related measures of vocabulary, such as new words in story retelling, sentence completion, and the Peabody Picture Vocabulary Test (PPVT [49]), and reading comprehension. The 14 studies included in the review contributed more than one effect size as a result of using multiple outcome measures or comparing more than one pair of conditions (i.e., multiple treatments and/or multiple comparison groups), resulting in a total of 60 effect sizes for the analysis. Of the 60 outcomes, 5 were on concepts about print, 13 were on phonological awareness, 39 on vocabulary, and 3 on reading comprehension (see Figure 1).

Four of the reviewed studies reported atypically large ES [20,21,48,51] if we compare them with the rest of the reviewed studies or previous research [12,13,14]. Ioannidis [54] points out that a single study cannot be seen in isolation and should be compared with the entire prior evidence on the same or similar questions. Another reason of atypically large ES, pointed by Ionnidis [54] could be a paradigm shift. In this regard, it is important to note that in the present review, most of the studies reporting large ES are in the “e-book vs. regular school program condition”. It could be inferred that children learning vocabulary with e-book interventions learn much more specific words than those not receiving any intervention at all. However, due to both random errors and biases, too large effect sizes require extra caution [54].

### 3.1. Participant Characteristics

In the 14 studies selected, a total of 1138 students were included with a mean age 63,92 months (with a mean age range between 52 and 82.2 months of age)—492 preschoolers, 507 kindergartens, and 139 first graders. The studies included 451 girls, 501 boys, and 186 students whose gender was not reported.

Across the studies, four papers [19,20,21,47] reported a total of 330 students at risk for learning disabilities. A total of 24 students who were English Language Learners (ELLs) were reported in one paper [42], and the number of low-SES children reported in one paper was 78 [51].

All selected studies were published in English, but conducted in different languages and countries: Six in Israel [19,20,24,41,47,51]; three in the US [42,44,45]); two in The Netherlands [48,50]; one in Canada [43]; one in South Africa [21]; and one in Taiwan [46].

### 3.2. Intervention Materials

The story books selected for the studies were both commercial e-books and e-books adapted by the researcher. In most cases, when authors selected a commercial e-book, their choice was based on a set of criteria, such as kindergarten teacher recommendations [41]; the inclusion of colorful pictures and a basic storyline appropriate for preschool and kindergarten children [42]; a good narrative structure and readability that provided descriptions of characters, settings, times, goals/initiating events, problems, and solutions/endings [24,44]; high-quality illustrations; ease of interaction (e.g., hot spots that were clearly identified); and interactive elements that did not interfere with the story text [39].

Specially constructed electronic versions were adapted by researchers to fit or create a story adequate to the cultural context [21] and to capture educational principles, especially those supporting literacy development. In some cases, children were offered a number of interactive modes of operation (i.e., read only, read story with dictionary, and read the story and play) [20] and different versions (i.e., with metacognitive guidance and the other without such guidance) [47]. Among the salient characteristics of some of the selected e-books was the “read story with dictionary”, a mode embedded in the e-book, which offers an oral reading of the text together with explanations of difficult words. To support individualized vocabulary acquisition, young readers could click the hotspots and access the respective screens as often as they wished. Each difficult word was enunciated clearly by the narrator as it appeared on the screen and was associated with pictures that support its meaning [20,47,51].

### 3.3. Literacy Measures

In the reviewed studies, reading knowledge was mainly assessed using measures, created by the researcher, of target text, words, or sounds instructed or included in the e-books selected for the respective studies. In many of the studies, the tasks were based on existing tests (i.e., adjusted for the research, such as a created picture vocabulary test, a frequently used test format for young children’s vocabulary learning) [24,41,44,45,46].

Measures created by the researcher, as opposed to norm-referenced standardized tests, are likely to be sensitive to growth in specific vocabulary [55]. As such, these measures are dependent on e-book text and should have greater outcomes than independent non-related e-book text measures. Additionally, interventions in the selected studies were not of a sufficient time duration to lead to changes in norm-referenced standardized tests or non-dependent measures of an e-book text.

The frequency of non-significant, small, medium, and large ES would be reported in each of the three studied conditions and on each literacy measure (see Figure 2 for a summary of the frequency of different ES in vocabulary and phonological awareness in the e-book and control conditions).

### 3.4. E-Book versus Print Book

Five studies compared e-books to print book in nine different conditions with a total of 17 reported outcomes (see Table 1). Two of the outcomes were on concepts about print, one with no significant effect and the other with a large effect of ES = 3.35 in favor of the e-book. Six of these outcomes were on phonological awareness with effects ranging from non-significant to small effect sizes (0.08 to 0.49) in favor of e-book use. Studies reported eight additional outcomes on vocabulary, one of them with negligible effects on students’ vocabulary, two of them with small-to-large effects (0.30 and 0.91) in favor of the print book use; in these two cases, the print book was read, and activities were guided by an adult, while the e-book condition was operated independently by the children. However, the other five outcomes on vocabulary in favor of the e-book use ranged from medium-to-large effect sizes (0.65 to 1.57). The conditions that favored the learning of vocabulary using the e-book versus the printed book used an enhanced e-book condition with vocabulary instructions versus listening to an adult-read print story with incidental vocabulary exposure [42] or using an e-book with scaffolding-like questions versus the experimenter reading aloud the print story one-on-one without receiving practice of the story [44]. In one of the conditions, the e-book was used only in the “read to me” mode versus the print book read by the experimenter without receiving any other practice; the outcomes favored the e-book [44]. Studies reported one outcome on reading comprehension in favor of the print book with a medium effect size, ES = 0.58.

### 3.5. Enhanced E-Book versus Non-Enhanced E-Book Conditions

Seven studies compared enhanced e-books to non-enhanced e-books in 12 different conditions and a total of 26 reported outcomes (see Table 2). Three of these outcomes were on phonological awareness, with effects ranging from non-significant to small size effects (0.15 to 0.48) in favor of enhanced conditions of e-book use, such as embedded metacognitive guidance in an educational e-book, which was found to support phonological awareness versus using an e-book without any metacognitive guidance [47]. Studies reported 21 outcomes on vocabulary, three of them with negligible effect sizes. Seven of them had small effect sizes ranging from ES = 0.26 to ES = 0.48. Four of them had medium effect sizes ranging from ES = 0.50 to ES = 0.74, and four of them had large effect sizes. All of them were in favor of the enhanced e-book conditions. Two of the reported outcomes concerning vocabulary were on the PPVT test [49] with non-significant effect sizes. The conditions that favored the learning of vocabulary were, for example, reading the e-book with adult vocabulary support versus dynamic visual vocabulary support embedded in the e-book [24]; visual vocabulary support versus no vocabulary support [24]; reading interactive animated e-books versus reading static e-books [48]; or embedded questions during the story versus embedded questions after the story [50]. Finally, two outcomes were reported on reading comprehension with large effect sizes, ES = 0.83 and ES = 0.9. The conditions that favored reading comprehension outcomes were teacher-delivered e-books with the 3R strategy “read, review, and recite” and keyword cues versus the same strategy without key words cues or the recite phase [46].

### 3.6. E-Book versus the Regular School Program

Six studies compared e-books with the regular school program in seven different conditions and a total of 17 reported outcomes (see Table 3). Three of these outcomes were reported on concepts about print, one of them with a negligible size; one with a medium size, and one with a large size. Four outcomes were reported on phonological awareness skills, one of them non-significant and three of them with medium size effects ranging from 0.52 to 0.54 in favor of the e-book condition. One of the conditions that favored the learning of phonological awareness skills showed children, when they clicked on a word, a large colorful frame displaying its phonetic segments in enlarged letters on the computer screen [19,51]. Studies reported 10 outcomes in vocabulary; eight of them had large effect sizes, in favor of the e-book. Conditions that favored the vocabulary outcomes were, for example, children listening to the audio of the story and engaged independently with the story activities, following verbal instructions that were part of the e-book [21], or children independently reading the static e-book versus playing computer games [48]. One outcome was in favor of the e-book condition with a medium size effect, and one non-significant outcome was reported for the PPVT test [49].

## 4. Discussion

This review provides a comprehensive assessment of a random control trial (RCT) and research evidence published in the last 10 years on the effectiveness of e-books on emergent literacy skills for children in preschool, kindergarten, and first grade. Our findings, although preliminary due to the limited scope of the available literature, provide researchers and practitioners with information for making evidence-based instructional decisions that can inform future research.

The primary objective of the present review was to examine the effectiveness of e-books for supporting early literacy skills, focusing on concepts about print, phonological awareness, vocabulary, and reading comprehension. Additionally, we explored the contributions of e-books compared to three conditions: Print books, e-books in non-enhanced conditions, and the regular school program.

With regard to the first aim on the effectiveness of e-book interventions for supporting literacy skills for young children, two outcomes out of five on concepts about print were found to produce large effects in favor of e-book interventions. Furthermore, modest gains were observed in the 13 outcomes on phonological awareness in favor of e-book conditions (see Figure 2). Additionally, out of 39 outcomes on vocabulary, 7 of them were found to produce small effect sizes, and 21 of them were found to produce medium-to-large effects; that is, 78% of outcomes in vocabulary clearly favored the e-book condition (see Figure 2). Finally, out of the 14 studies, only three outcomes were reported on reading comprehension, and two of them were found to produce large size effects in favor of e-book interventions. In considering the effectiveness of e-books for supporting literacy skills in young children, we would like to highlight three key conclusions.

First, as in previous reviews [8,11] and meta-analyses [12,13], we found evidence that e-book interventions have a significant positive effect on measures of vocabulary when used by young readers compared to print books and regular school programs. Vocabulary performance was also higher in the enhanced e-book condition versus the non-enhanced e-book one. Among the key characteristics of e-book reading that could have a positive effect on vocabulary learning is research attention to the interactive dictionary feature accessed through hotspots (prompt access to words pronunciation and meanings) emerged as a factor that strengthens word meaning [20,24,50].

Second, as in a previous review [11], we found that the reading of e-books led to modest gains in phonological awareness in favor of the e-book condition.

Third, due to a limited number of studies addressing concepts about print and reading comprehension (see Figure 3), no estimates of the practical effect could be presented for these abilities, although positive effects in favor of the e-book are found in both literacy skills.

Concerning the contributions of e-book interventions compared to print books for supporting early literary skills, out of the five reviewed studies, four were found to produce small-to-large size effects in favor of the e-book condition. As in a previous meta-analysis [12], we found that interactive e-book stories showed an advantage of the multimedia element over children encountering traditional print books without receiving adult support. This finding, reported in the reviewed studies [20,44], might suggest that the multimedia features provide similar scaffolding to that of an adult for children’s literacy experiences. Interactive e-books with audio narration allow children to learn vocabulary [20,44] and become aware of sounds [20,41], indicating that using e-books independently may be a valuable practice for promoting children’s acquisition of emergent literacy skills without adult assistance when teachers or parents are otherwise occupied. However, previous research concluded that learning improves when the e-book is shared with an adult [8,9,10,11]; digital literacy activities, such as e-book reading, could ameliorate the lack of adult support during story reading [44] and give educators and parents extra time, which may be needed in the context of distress or crisis, such as the current pandemic situation.

Similarly, all seven reviewed studies that compared enhanced e-book reading interventions to e-book non-enhanced conditions reported non-significant to large effect sizes in favor of enhanced e-book interventions. The e-books selected in the enhanced conditions of the reviewed studies presented interactive characteristics, illustrations, automatic animations, and hotspots that are activated by touching or pressing the touch screen to generate sound and animation [24,42,44,45]. Consequently, the selected e-books are easily accessible for emergent readers who cannot yet decode text or are just beginning to learn to decode it in comparison with non-enhanced, non-interactive, or static e-books. Well-designed and effective educational e-books for young children can serve as a good opportunity to enhance children’s language and literacy [51].

Finally, the six studies comparing e-book interventions versus regular school programs reported quite favorable outcomes of e-book intervention. However, while preschool staff are aware of ways in which literacy might be supported, the learning that can ensue from interactions with technology has not yet been fully recognized, and little attention has been paid to the ways in which children’s emerging competences with technology are supported and can be ameliorated [56]. In fact, of the 14 selected studies for the present review, only one of them [46] is delivered by teachers. Educators and parents are concerned about the perceived negative effects of digital media on children’s health [1]. Nonetheless, guidance on the use of the e-book in schools and at home may be more helpful than restrictions and warnings regarding the risk of young children’s exposure to digital activities and screens.

To address the last aim of the present study, we discuss some aspects of the reviewed studies that are worth mentioning. The first is related to special populations. In the present review, special attention was given to the effects of e-book interventions for the different groups of children by testing every effect separately for disadvantaged and non-disadvantaged children. We found that children at risk of learning disabilities [19,20,21,47], ELL [42], and low SES [51] received a clear positive impact from e-book interventions in all the studied literacy abilities. It is plausible that for children who do not fully understand the story of a book because they lack the language and comprehension skills necessary, non-verbal information from animations and sound effects can bridge the gaps [13]. Additionally, the “app gap” has shrunk substantially, and more children now have access to digital devices [2]; this could create opportunities for children from different backgrounds in online learning conditions, which, for example, has been the case in the present crisis situation.

A second aspect is related to the quality of e-book design and appropriate selection. Although today, e-books, on one hand, have a more educational value for young children compared to those that were on the market a decade ago [16], on the other hand, older types of digital books, which include sound, animations, and interactive hotspots, unlike modern e-books, are operated by a mouse on a desktop computer, which is much less intuitive than modern tablets where children can use their fingers on a touch screen [4]. However, regarding these recent improvements in e-book design, educators and parents need advice about principles of good educational e-books for young children. E-book analyses are needed to make recommendations on what comprises good e-books for young children in order to better serve designers, parents, educators, and children [16].

A third aspect worth mentioning is related to adult implications in e-book interventions. As stated above, the literature base offers preliminary evidence suggesting that adult support may provide additional benefits beyond the interactive features in e-books [8,9,10,11]. However, previous research suggests that young children acquire a wide range of competencies when interacting with technology, which are developed in ways that are not necessarily the result of direct teaching [57]. Consequently, the adult role may move away from direct support while using the e-book to a more distal role, centering on selection and encouragement [3] and the design of appropriate strategies [46] and contexts [21] to learn from e-books.

The fourth aspect refers to the possible negative effects of digital media on children’s health. Internet Gaming Disorder and Gaming Disorder have been included as conditions in the DSM-5 [33] and the ICD-11 [32] respectively, justifying more clinical research and experience on this topic. As Wolf remarks, the reality is that we cannot go back; nor should we move ahead thoughtlessly. In this sense research conducted by E-READ network, New America, the Joan Ganz Cooney Center, and the MacArthur Foundation program encourage educators to help the habits that spur learning no matter where the text comes from, no matter whether the image is on paper or screen [58].

Finally, we would like to consider the limited number of intervention sessions of the reviewed studies, ranging from 2 to 16 sessions. In this sense, one of the selected studies [51] found that the contribution of repeated reading (3 vs. 6 repetitions) contributed to greater vocabulary gains for preschool and kindergarten children after six repetitions compared to three repetitions. Similarly, only one of the reviewed studies [21] presents a follow up of the outcomes. Although the present research findings suggest that reviewed e-book interventions can be effective to increase phonological awareness and vocabulary outcomes among young children, the extent to which gains are generalized and maintained over time has been, at best, scarcely examined. Studies with more intervention and follow-up sessions are needed to better understand the effectiveness of e-book interventions.

Currently, children across all levels of society are using digital technologies and mobile media on a daily basis [1,2,34]. However, research on the educational value of digital literacy for children is still in its early stages. We are only beginning to understand what this digital shift means for young children’s literacy development [3]. However, there is evidence of how children can improve their emergent literacy skills using e-books. Carefully chosen e-books presented on a tablet can offer a highly independent learning experience for young children. As technology becomes increasingly ubiquitous, practitioners and parents must be guided, not only on the frequency and content of the media their child uses, but also on how to introduce media at schools and at home in dedicated spaces and at set times, and how to initiate social and creative ways to use digital media—all of this must take place without displacing their child’s basic needs [34].

With the current health crisis, we have learned that it is possible and highly probable that distance learning involving schools and families assisting children will play a key role in education in the future. Reading is a key academic and life skill that should be encouraged from the early stages of life; not all young children have the opportunity to attend school but can use e-books and digital devices to develop literacy [21]. Therefore, parents and practitioners need a better understanding on how e-books, and what type of e-books, could contribute to the development of emergent literacy skills in young children, taking into account the recommendations given by international organizations to avoid health issues in children [25].

## 5. Conclusions

In the present research synthesis, including 14 studies and 1138 children aged 7 years or younger, we found evidence that e-book stories improve phonological awareness and vocabulary as compared to traditional stories and regular school programs. Additionally, well-selected, animated, and interactive e-books presented in contexts and situations carefully designed by researchers showed better outcomes in literacy skills than older types of static digital books. The addition of enhanced conditions in the software and selection of the e-book, as well as the systematic adult planning of intervention sessions, resulted in greater intervention effects than non-enhanced conditions.

Young children can listen to storybooks not only when an adult reads to them from a printed version, but also by themselves using e-books on computers and tablets. Results of the present study show that the presence of an adult does not have an advantage for phonological awareness and vocabulary learning. We are aware that young children learn best from exchanges with caring adults, but in the case of learning with e-books, their role might shift away from direct support to a more distal role while children are using the e-book (i.e., adequate selection of the e-book, the design of strategies, and the adaptation to the context where the e-book is used).

Children living in a deprived context, at risk of learning disabilities, and English Language Learners benefited from all the reviewed e-book interventions, which highly improved their literacy skills, regarding concepts about print, phonological awareness, vocabulary, and reading comprehension. Digital learning and e-book reading represent a potential compensatory strategy for these children.

Although we found positive results in favor of e-book interventions, the present study findings should be interpreted with caution due to the limited scope of the available literature, the limited number of intervention sessions, as well as the lack of follow-up studies to test if e-book interventions produce lasting and generalizable skill gains.

However, in the current crisis situation, online learning has increased in all countries, and practitioners and families should receive advice on quality e-books that could be used with young children and on how to organize appropriate learning environments using e-books. The goal is to continue to promote literacy in this key state of development, which is related to emergent reading and reading achievement, even in the most adverse contexts and situations.

The findings of the present study should be interpreted with caution due to a limited number of studies included. In order to better understand the impact of e-book on children’s emergent literacy skills, future research on e-book interventions must (a) involve schools and families from socio-demographically diverse areas and (b) implement study designs over longer periods of time.

## Figures and Tables

**Figure 1 ijerph-18-06510-f001:**
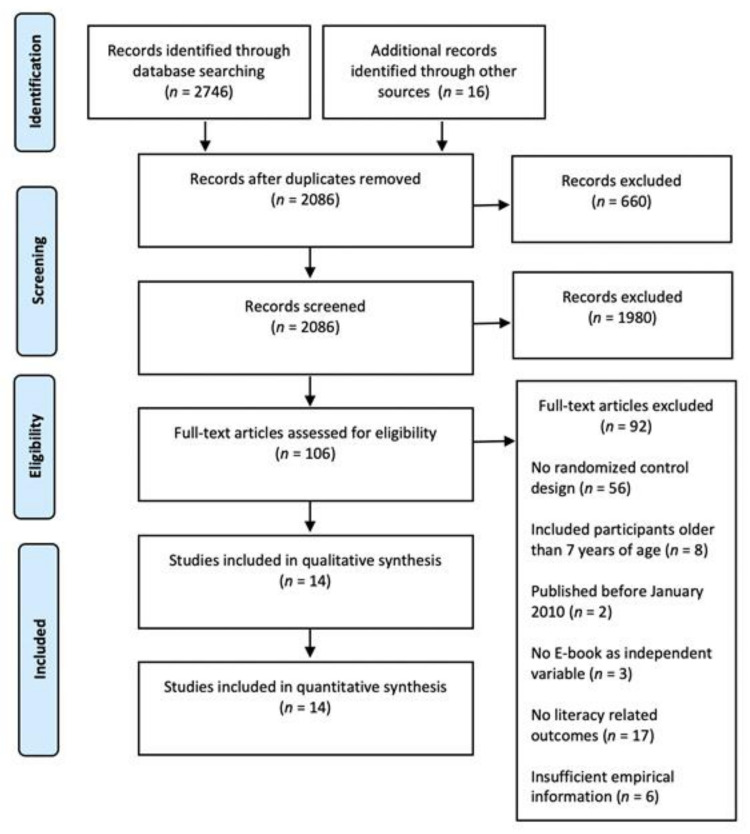
Prisma chart describing the search and inclusion procedures [39].

**Figure 2 ijerph-18-06510-f002:**
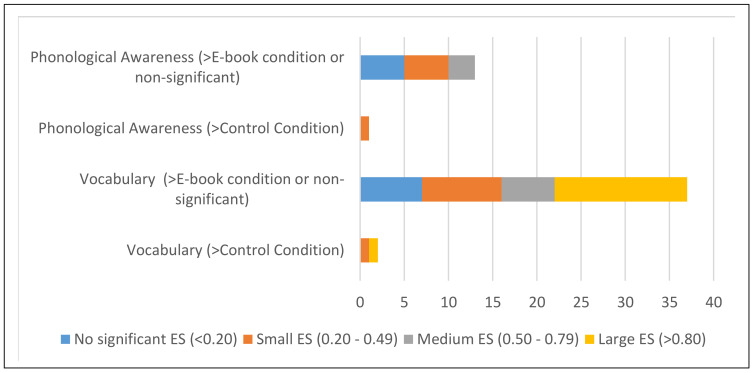
Summary of the frequencies of ES (effect size) in vocabulary and phonological awareness in the e-book and control conditions.

**Figure 3 ijerph-18-06510-f003:**
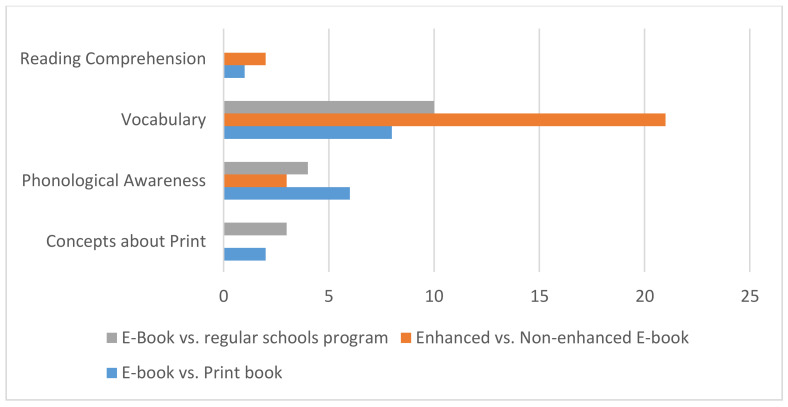
Number of literacy outcomes in the three compared conditions.

**Table 1 ijerph-18-06510-t001:** Study information (e-book versus print book).

Article Info	Study Design	Participant Information	Mean Age (m)	Intervention Description	Number and Type of Books	Intervention Duration	Outcomes Effect Sizes Calculated Using Carlson and Schmidt [40]
[41]	TC	*N* = 50 kindergarten children Treatment T1 and Control C1 SES not specified Normal verbal intelligence	66	Delivered by experimenter (on computers) T1: Children independently listened to a recording e-book format. Activities guided by a recorded explanation (*n* = 25). C1: Children listened to a print book story read by an experimenter. Activities guided by the experimenter (*n* = 25).	1 Adapted E-Book 1 Print Book (same story)	5 School Sessions 10–30 min.	CP: T1 = C1 (es = 0.07) PA: T1 > C1 (es = 0.30) EV: T1 < C1 (es = 0.91) RC: T1 < C1 (es = 0.58)
[42]	ST	*N* = 24 kindergarten children Treatment T1 and Control C1: Low SES English Language Learners	66.9	Delivered by experimenter (on computers) T1: A technology-enhanced shared e-book reading with vocabulary instruction embedded in prerecorded audio files and adult mediation. C1: Children listened to an adult-read print story book with incidental vocabulary exposure.	4 Commercial E-Books 4 Commercial Print Books (same stories)	6 Summer School Program Sessions	RV: T1 > C1 (es = 0.82) EV: T1 > C1 (es = 1.31)
[20] ^1^	MT	*N* = 110 preschool children in all treatment conditions Treatment Conditions: (T1), (T2) and (T3) Condition 1: T1 and T2 Middle SES Children at risk for learning disabilities	72	Delivered by experimenter (on computers) T1: Children read and worked independently with the e-book after they were shown how the software operated (*n* = 42). T2: The experimenter read the printed book to children (*n* = 34).	1 Adapted E-Book (3 modes of operation) 1 Print Book (same story)	6 School Sessions 20–35 min.	CP: T1 > T2 (es = 3.35) PA: T1 > T2 (es = 0.49) EV: T1 > T2 (es = 0.65)
[43]	MT	*N* = 94 kindergarteners in all treatment conditions Treatment Conditions: (T1), (T2) and (T3) Condition 1: T1 and T2 SES not specified	53	Delivered by experimenter (on iPad tablet) T1: The experimenter read the alphabet e-book aloud, and then the children could explore the e-book (*n* = 33). T2: The experimenter read the alphabet print book aloud, and then the children could explore the print book (*n* = 30).	Several Commercial Alphabet E-Books And Several Alphabet Print Books	16 School Sessions 20 min.	Letter name: T1 = T2 (es = 0.12) Letter sound: T1 < T2 (es = 0.27)
[43]	MT	Condition 2: T1 and T3 SES not specified	53	Delivered by experimenter (on iPad tablet) T1: The experimenter read the alphabet e-book aloud, and then the children could explore the e-book (*n* = 33). T3: The experimenter read the story print book aloud, and then the children could explore the print book (*n* = 29).	Several Alphabet E-Books and Several Alphabet Print Books	16 School Sessions 20 min.	Letter name: T1 = T3 (es = 0.03) Letter sound: T1 = T3 (es = 0.08)
[44] ^2^	MT	*N* = 72 preschool children in all treatment conditions Treatment Conditions: (T1), (T2), (T3) and (T4) Condition 1: T1 and T2	53	Delivered by experimenter (on iPad tablet) T1: Children were shown how the software operated and interacted individually with the story with scaffolding-like questions (*n* = 36). T2: The experimenter read aloud the print story one-on-one as well as using scaffolding-like questions (*n* = 36).	1 Commercial E-Book 1 Print Book (same story)	2 School Sessions	RV: T1 = T2 (es = 0.09)
[44]		Condition 2: T3 and T4	53	T3: Children were shown how the software operated and interacted individually with the story in the “read to me” mode (*n* = 36). T4: The experimenter read aloud the print story one-on-one without receiving practice of the story (*n* = 36).	1 Commercial E-Book 1 Print Book (same story)	2 School Sessions	RV: T3 > T4 (es = 1.37)
[44]		Condition 3: T1 and T4	53	T1: Children were shown how the software operated and interacted individually with the story with scaffolding-like questions (*n* = 36). T4: The experimenter read aloud the print story one-on-one without receiving the practice story (*n* = 36).	1 Commercial E-Book 1 Print Book (same story)	2 School Sessions	RV: T1 > T4 (es = 1.57)
[44]		Condition 4: T3 and T2 SES not specified	53	T3: Children were shown how the software operated and interacted individually with the story in the “read to me” mode (*n* = 36). T2: The experimenter read aloud the print story one-on-one as well as using scaffolding-like questions (*n* = 36).	1 Commercial E-Book 1 Print book (same story)	2 School Sessions	RV: T3 < T2 (es = 0.30)

Note. TC = treatment vs. condition; MT = multi-treatment groups; ST = single treatment group. T1 = treatment 1; T2 = treatment 2; T3 = treatment 3; T4 = treatment 4; C1 = control 1; SES = socioeconomic status; CP = Concepts about Print; PA = Phonological Awareness; EV = Expressive Vocabulary; RV = Receptive Vocabulary; RC = Reading Comprehension. ^1^ Condition 2 (T1) and (T3) of this study are reported in Table 3. ^2^ Condition 5 (T1) and (T3) of this study are reported in Table 2.

**Table 2 ijerph-18-06510-t002:** Study information (enhanced e-books versus non-enhanced e-books).

Article Info	Study Design	Participant Information	Mean Age (m)	Intervention Description	Number and Type of Books	Intervention Duration	Outcomes Effect Sizes Calculated Using Carlson and Schmidt [40]
[45]	TC	*N* = 30 preschool children: Treatment T1 and Control C1: High, Middle, and Low SES	55	Delivered by experimenter (on iPad tablet) T1: Children read digital interactive focal e-book three times, while the experimenter provided cues for attention and praise for on-task behavior when necessary (*n* = 15). C1: Children read digital non-interactive storybooks three times, while the experimenter provided cues for attention and praise for on-task behavior when necessary (*n* = 15).	1 Commercial E-Book (2 modes of operation)	3 School Sessions	RV: T1 = C1 (es = 0.01) EV: T1 > C1 (es = 0.26)
[24]	MT	*N* = 144 preschool children in all treatment conditions Treatment Conditions: (T1), (T2), (T3) and (T4) Condition 1: (T1 and T2)	60	Delivered by experimenter (on iPad tablet) T1: Children read the focal e-book three times with adults’ vocabulary support (*n* = 37). T2: Children read the focal e-book three times with dynamic visual vocabulary support (*n* = 32).	1 Adapted E-Book (4 modes of operation)	3 School Sessions 20–25 min.	RV: T1 > T2 (es = 0.46) EV: T1 > T2 (es = 0.69) New words in story retelling: T1 > T2 (es = 1.07)
[24]		Condition 2: (T3 and T4)	60	T3: Children read the focal e-book three times with static visual vocabulary support (*n* = 38). T4: Children read the focal e-book three times without vocabulary support (*n* = 37).	1 Adapted E-Book (4 modes of operation)	3 School Sessions 20–25 min.	RV: T3 > T4 (es = 0.23) EV: T3 > T4 (es = 0.51) New words in story retelling: T3 = T4 (es = 0.07)
[24]		Condition 3: (T2 and T3) Middle SES	60	T2: Children read the focal e-book three times with dynamic visual vocabulary support (*n* = 38). T4: Children read the focal e-book three times without vocabulary support (*n* = 37).	1 Adapted E-Book (4 modes of operation)	3 School Sessions 20–25 min.	RV: T2 > T4 (es = 0.36) EV: T2 > T4 (es = 0.74) New words in story retelling: T2 > T4 (es = 0.50)
[46]	MT	*N* = 74 first graders children in all treatment conditions Treatment Conditions (T1), (T2), and (T3) Condition 1: (T1 and T2)	82.2	Delivered by teacher (on computer) T1: Teacher used the 3R (read, review, and recite) strategy and keyword cues (*n* = 26). T2: Teacher used the 3R (read, review, and recite) strategy without keyword cues (*n* = 23).	1 Commercial E-Book	3 School Sessions	PA: T1 = T2 (es = 0.15) RC: T1 > T2 (es = 0.83)
		Condition 2: (T1 and T3) SES not specified	82.2	T1: Teacher used the 3R (read, review, and recite) strategy and keyword cues (*n* = 26). T3: Teacher read the e-book twice, without the recite phase or any guidance instruction (*n* = 25).			PA: T1 > T3 (es = 0.41) RC: T1 > T3 (es = 0.91)
[47] ^1^	MT	*N* = 78 kindergarten children in all treatment conditions Treatment Conditions (T1), (T2), and (T3) Condition 1: (T1 and T2) SES not specified Children at risk for learning disabilities	52	Delivered by experimenter (on computer) T1: Children independently read the e-book with metacognitive guidance (*n* = 26). T2: Children independently read the e-book without metacognitive guidance (*n* = 26).	1 Adapted E-Book (2 versions and 3 modes of operation)	6 School Sessions 20 min.	PA: T1 > T2 (es = 0.48) RV: T1 = T2 (es = 0.09)
[48] ^2^	MT	*N* = 136 kindergarten children in all treatment conditions Treatment Conditions (T1), (T2), (T3) and (T4) Condition 1: (T1 and T2)	63.5	Delivered by experimenter (on a computer) T1: Children independently read the interactive animated e-book (*n* = 33). T2: Children independently read the non-interactive animated e-book (*n* = 36).	5 Commercial E-Books (3 modes of operation)	8 School Sessions 12–14 min.	PPVT [49] T1 = T2 (es = 0.09) EV: T1 > T2 (es = 0.88)
		Condition 2: (T2 and T3) SES not specified	63.5	T1: Children independently read the interactive animated e-book (*n* = 33). T3: Children independently read the static e-book (*n* = 33).	5 Commercial E-Books (3 modes of operation)	8 School Sessions 12–14 min.	PPVT [49] T1 = T3 (es = 0.06) EV: T1 > T3 (es = 5.74)
[50]	ST	*N* = 20 Kindergarten children in all treatment conditions Experiment 1: (T1 and T2) SES not specified	54.5	Delivered by experimenter (on computer) T1: A computer aid introduced the story and children independently read a series of e-books with embedded multiple-choice questions during the story book. T2: A computer aid introduced the story and children independently read a series of e-books with embedded multiple-choice questions after the story books.	5 Commercial E-Books (3 modes of operation)	5 School Sessions	RV: T1 > T2 (es = 0.26) EV: T1 > T2 (es = 0.48)
[50]		*N* = 27 Kindergarten children in all treatment conditions Experiment 2: (T1 and T2) SES not specified	57,56	Delivered by experimenter (on computer) T1: Children independently read a series of e-books with embedded multiple-choice questions. T2: Children independently read a series of e-books with hotspots that included word meaning explanation.	5 Commercial E-Books (3 modes of operation)	5 School Sessions	RV: T1 < T2 (es = 0.42) EV: T1 > T2 (es = 1.67)
[44] ^3^	MT	*N* = 72 preschool children in all treatment conditions Treatment Conditions: (T1), (T2), (T3), and (T4) Condition 5: T1 and T3 SES not specified	53	Delivered by experimenter (on iPad tablet) T1: Children were shown how the software operated and interacted individually with the story with scaffolding-like questions (*n* = 36). T3: Children were shown how the software operated and interacted individually with the story in the “read to me” mode (*n* = 36).	1 Commercial E-Book	2 School Sessions	RV: T1 > T3 (es = 0.25)

Note. TC = treatment vs. condition; MT = multi-treatment groups; ST = single treatment group. T1 = treatment 1; T2 = treatment 2; T3 = treatment 3; T4 = treatment 4; C1 = control 1; SES = socioeconomic status; CP = Concepts about Print; PA = Phonological Awareness; EV = Expressive Vocabulary; RV = Receptive Vocabulary; RC = Reading Comprehension. PPVT = Peabody Picture Vocabulary Test; ^1^ Condition 3 (T1) and (T3) of this study are reported in Table 3. ^2^ Condition 3 of this study is reported in Table 3 (T3) and (T4). ^3^ Other conditions of this study presented in Table 1.

**Table 3 ijerph-18-06510-t003:** Study information (e-book versus the regular school program).

**Article Info**	**Study Design**	**Participant Information**	**Mean Age (m)**	**Intervention Description**	**Number and Type of Books**	**Intervention Duration**	**Outcomes Effect Sizes Calculated Using Carlson and Schmidt [40]**
[21]	TC	*N* = 65 first graders children Treatment T1 and Control C1: Low SES at risk for reading failure	79	Delivered by experimenter (on android tablet) T1: Children listen to the audio of the story and engaged independently with the story and activities, following verbal instructions that were part of the e-book (*n* = 33). C1: Regular school program (*n* = 32).	1 Adapted E-Book	6 School Sessions 20 min.	RV: T1 > C1 (es = 2.05) Sentence Completion: T1 > C1 (es = 5.01) EV: T1 > C1 (es = 1.52)
[51]	TC	*N* = 78 kindergartners Treatment T1 and Control C1: Low SES	66	Delivered by experimenter (on computer) T1: Children read independently from an e-book that automatically provided the meaning of difficult words in the story (*n* = 40). C1: Regular kindergarten program (*n* = 38).	1 Adapted E-Book	6 School Sessions	RV: T1 > C1 (es = 1.67) EV: T1 > C1 (es = 2.02) Word Production: T1 > C1 (es = 1.70)
[19]	TC	*N* = 76 preschool children Treatment T1 and Control C1: At risk for learning disabilities SES not specified	71.2	Delivered by experimenter (on computer) T1: Children read independently from an e-book and then the experimenter asked some questions about the story (*n* = 42). C1: Regular preschool program (*n* = 34).	1 Commercial E-Book	6 School Sessions 20–35 min.	CP: T1 > C1 (es = 0.50) PA: T1 > C1 (es = 0.54)
[19]	TC	*N* = 60 preschool children Treatment T1 and Control C1: Typically developing children SES not specified	71.2	Delivered by experimenter (on computer) T1: Children read independently from an e-book and then the experimenter asked some questions about the story (*n* = 28). T2: Regular preschool program (*n* = 32).	1 Adapted E-Book (3 modes of operation)	6 School Sessions 20–35 min.	CP: T1 > C1 (es = 0.09) PA: T1 > C1 (es = 0.52)
[20] ^1^	MT	*N* = 110 preschool children in all treatment conditions (T1), (T2), and (T3) Condition 3: T1 and T3 At risk for learning disabilities Middle SES	72	Delivered by experimenter (on computers) T1: Children read and worked independently with the e-book after they were shown how the software operated (*n* = 42). T3: Regular preschool program (*n* = 34).	1 Adapted E-Book (3 modes of operation)	6 School Sessions 20–35 min.	CP: T1 > T3 (es = 3.13) PA: T1 > T3 (es = 0.54) EV: T1 > T3 (es = 1.93)
[47] ^2^	MT	*N* = 78 kindergarten children in all treatment conditions Treatment Conditions (T1), (T2), and (T3) Condition 2: (T1 and T2) SES not specified Children at risk for learning disabilities	52	Delivered by experimenter (on computer) T1: Children independently read the e-book with metacognitive guidance (*n* = 26). T3: Regular kindergarten program (*n* = 26).	1 Adapted E-Book (2 versions and 3 modes of operation)	6 School Sessions 20 min	PA: T1 > T3 (es = 0.14) RV: T1 > T3 (es = 0.63)
[48] ^3^	MT	*N* = 136 kindergarten children in all treatment conditions Treatment Conditions (T1), (T2), (T3) and (T4) Condition 3: (T3 and T4) SES not specified	63.5	Delivered by experimenter (on computer) T3: Children independently read the static e-book (*n* = 38). T4: Children play computer games (*n* = 34).	5 Commercial E-Books	8 School Sessions 12–14 min.	PPVT [43]: T1 > T4 (es = 0.14) EV: T1 > T4 (es = 4.32)

Note. TC = treatment vs. condition; MT = multi-treatment groups; T1 = treatment 1; T2 = treatment 2; T3 = treatment 3; T4 = treatment 4; C1 = control 1; SES = socioeconomic status; CP = Concepts about Print; PA = Phonological Awareness; EV = Expressive Vocabulary; RV = Receptive Vocabulary; RC = Reading Comprehension. PPVT = Peabody Picture Vocabulary Test; ^1^ Other conditions of this study reported in Table 1. ^2^ Other conditions of this study reported in Table 2. ^3^ Other conditions of this study reported in Table 2.

## Data Availability

Data available on request from the corresponding author.

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
