# Peer review of "The Impact of E-Book Reading on Young Children’s Emergent Literacy Skills: An Analytical Review"

_ijerph, 2021, doi:10.3390/ijerph18126510_

Round 1

Reviewer 1 Report

Overall, this is a timely and relevant manuscript.  The findings,  in light of the most recent COVID-19 pandemic and online learning for students, are especially poignant as families have had to adapt to learning via digital platforms.  Guidance on how to do this more effectively is much needed and evidenced in this manuscript.  

A couple of edits that might improve readability over all:

--when using i.e., e.g., or other Latin abbreviations, it is customary to use them in parentheses (see p. 2, lines 81-82). 

--p. 3, line 137: list grades in ascending order - preschool, kindergarten and grade 1.

Gersten et al. (2005) offer measures of quality indicators for meta-analyses and systematic reviews.  This would be an interesting aspect to add to your manuscript, in addition to analyses on studies that meet the criteria for high quality vs. low quality.  The paper looks great as is, this would just be something that could strengthen your findings.

The paper is well written and thoroughly describes the process used for analysis of the included papers.  The findings contribute valuable insight into the role of e-books, as well as parent involvement and repeated reading to improved vocabulary - all of which are needed as we delve further into the digital world.

One reference that might have been useful to include is Maryanne Wolf's Reader, Come Home.  In it, she discusses the impact of digital influences on the deep reading that students do with actual books versus digital media.  

Author Response

Reviewer #1’s comments

Dear Reviewer#1:

We like to thank you for the suggestions to improve the manuscript. Please, below you will be able to find our replies to all your proposals.

A couple of edits that might improve readability over all:

--when using i.e., e.g., or other Latin abbreviations, it is customary to use them in parentheses (see p. 2, lines 81-82)

Taking this comment into account, in page 2, lines 83-85, parentheses have been added

--p. 3, line 137: list grades in ascending order - preschool, kindergarten and grade 1.

Taking this comment into account, in page 3, line 146, grades are listed in ascending order

Gersten et al. (2005) offer measures of quality indicators for meta-analyses and systematic reviews.  This would be an interesting aspect to add to your manuscript, in addition to analyses on studies that meet the criteria for high quality vs. low quality.  The paper looks great as is, this would just be something that could strengthen your findings.

Thank you for the reference which we found very valuable. We have not used it on this occasion so as not to lengthen the article excessively.

One reference that might have been useful to include is Maryanne Wolf's Reader, Come Home.  In it, she discusses the impact of digital influences on the deep reading that students do with actual books versus digital media.  

Thank you for the reference. We have checked it out and found it very relevant to our work. We have added a citation on page 17, lines 787-794.

Reviewer 2 Report

The major falw of the paper is that is out of the scope of this Journal.

The manuscript fails to explain how e-books relate to health? Is the focus on literacy (e-literacy through e-books or promoting health?) Is 'health' referring to some form of 'well-being'? If so, authors should check the Leuven Involvement Scale and cross-check their findings.

What does emergent literacy has to do with Environmental Research and Public Health? I think authors need to re-think and perhaps ´re-comceptualise' the study, most likely removing the sections on health and leaving what strictly related to emergent literacy (how emergent literacy can be promoted through e-books).

In additon, the manuscript is too lenghty and discourages the reader to follow the work.

Author Response

Dear Reviewer#2:

We like to thank you for the suggestions to improve the manuscript.

Screen use by children has increased and starts at an early age. Potential effects of screen use on children`s health and development has been well documented (see references in the manuscript, page 3, lines 112-115)

In this regard, the International Classification of Disease (ICD-11), as well as, the Diagnostic and Statistical Manual of Mental Disorders 5 (DSM-5) have included conditions called “game disorder” and “internet game disorder” respectively. We have added this information to the manuscript (please, see page 3, lines 116-122). E-books also have increasingly sophisticated games.

Although research on the effects of digital devices on children's health is still limited, we feel it is necessary to highlight the fact that there may be undesirable effects on health from the excessive or inadequate use of digital devices by young children.

About the Leuven Scale for early years, emotional well-being, as well as, levels of involvement in children might be negatively affected by inadequate use of digital devices.

Emergent Literacy is not related to Public Health in a broad sense, but the use of digital devices by young children is.

Reviewer 3 Report

Regarding the content, I do not have any changes to recommend, it makes a good literary review to support the relevance of the problem to be studied and a good structuring of the content, it uses the correct methodology for this type of study and it is a consistent and well-detailed methodology to give significance to the results they show, makes a good discussion of the results with respect to the studies carried out previously, and marks the conclusion obtained well. Although some references to review methodologies should be included in ‘Materials and Methods’ section.

The format of tables 1 and 2 must be corrected: there are rows without lines that separate them and the header is repeated in the middle of the table (the table header can only be repeated if the table has more than one page, where the header of the table should go to the top of the page).

And in the ‘Conclusions’ section, you should add a paragraph at the end that talks about the limitations and includes deductions for future research.

In the ‘Author Contributions’ section, names must be initials only.

And in the ‘References’ section, the volume, number and pages of the journal articles should be put according to the template. Example: 5 (1), 11-20 ---- would be: Volume 5, Number 1, pp. 11-20. Review this section by following the template.

Once the authors have made these changes, I think the article is publishable.

Author Response

Dear Reviewer#3:

We like to thank you for the suggestions to improve the manuscript. Please, below you will be able to find our replies to all your proposals.

Although some references to review methodologies should be included in ‘Materials and Methods’ section.

Taking this comment into account, in page 4, lines 192-194, a reference to review methodologies has been added.

The format of tables 1 and 2 must be corrected: there are rows without lines that separate them and the header is repeated in the middle of the table (the table header can only be repeated if the table has more than one page, where the header of the table should go to the top of the page).

Taking this comment into account, tables 1 and 2 were corrected.

And in the ‘Conclusions’ section, you should add a paragraph at the end that talks about the limitations and includes deductions for future research.

Taking this comment into account, a paragraph in the conclusions section has been added. See page 18, lines 862-866.

In the ‘Author Contributions’ section, names must be initials only.

Names are initials only

And in the ‘References’ section, the volume, number and pages of the journal articles should be put according to the template. Example: 5 (1), 11-20 ---- would be: Volume 5, Number 1, pp. 11-20. Review this section by following the template.

In the Reference section, instructions for authors have been followed

https://www.mdpi.com/journal/ijerph/instructions

Reviewer 4 Report

The article is well presented and adds to the field by focusing on recent studies and restricting results to RCTs only. The consequence is that there are simply not enough studies to conduct a "proper" meta-analysis. At the same time, I believe that the authors could do more to use meta-analytic techniques - especially in relation to the large fluctuations in effect sizes (that were above 1, 2, and 3) the distribution does not make sense and calls for more c careful treatment explained before. It seems the authors went to rely on the frequency of effects instead of size but such a decision needs to be explicit.

In the results section, I call on the authors to create a graphic representation that would make the reporting in print clearer.

Avoid using negative effect sizes in text just let the text indicate the direction.

Effect size of change are hard to interpret and there needs to be an examination of potential floor and ceiling effects that may be the reason for such large effect sizes.

Clarify early on that the reason you did not conduct a meta-analysis is that there were too few studies- just say it.

Inter-rater agreement should use a more sophisticated metric such as Kappa.

Comma use is the main grammatical issue. .g. Lines 130-132

P 1. Line 27 I would suggest using children instead of infants

P. 10 line 239 needs to add a - after post

P. 10 Line 272 you must decide if you are using commas  orperiods for decimals (consult the guide by the journal)

Author Response

Dear Reviewer#4:

We like to thank you for the suggestions to improve the manuscript. Please, below you will be able to find our replies to all your proposals.

At the same time, I believe that the authors could do more to use meta-analytic techniques - especially in relation to the large fluctuations in effect sizes (that were above 1, 2, and 3) the distribution does not make sense and calls for more c careful treatment explained before. It seems the authors went to rely on the frequency of effects instead of size but such a decision needs to be explicit.

Taking this comment into account, a comment has been added in page 14, lines 570-572.

In the results section, I call on the authors to create a graphic representation that would make the reporting in print clearer.

Taking this comment into account, two graphics have been designed (see Figure 1 and Figure 2)

Avoid using negative effect sizes in text just let the text indicate the direction.

Negative signs of effect sizes have been deleted

Effect size of change are hard to interpret and there needs to be an examination of potential floor and ceiling effects that may be the reason for such large effect sizes.

Taking this comment into account and to justify ES change and large ES, a paragraph has been added (please see page 12, lines 466-475)

Clarify early on that the reason you did not conduct a meta-analysis is that there were too few studies- just say it.

Taking this comment into account a comment has been added in page 4, line 183

Inter-rater agreement should use a more sophisticated metric such as Kappa.

Both percent agreement and kappa have strengths and limitations. If there is likely to be much guessing among the raters, it may make sense to use the kappa statistics, but if the raters are well trained and little guessing is likely to exist, the researcher may safely rely on percentage agreement to determine interrater reliability (McHugh, 2012).

In the case of the present study guessing is unlikely as very specific data is sought.

McHugh, M.L. (2012). Interrater reliability: the kappa statistic. Biochem Med, 22(3), 276-282

Comma use is the main grammatical issue. .g. Lines 130-132

Done

P 1. Line 27 I would suggest using children instead of infants

Done

  1. 10 line 239 needs to add a - after post

Done

  1. 10 Line 272 you must decide if you are using commas  orperiods for decimals (consult the guide by the journal)

We have decided to use periods.